# Vision-and-Language Training Helps Deploy Taxonomic Knowledge but Does Not Fundamentally Alter It

**Yulu Qin,**[1,*] **Dheeraj Varghese,**[2,*] **Adam Dahlgren Lindström,**[3] **Lucia Donatelli,**[4]
**Kanishka Misra**,[5,6,†] and **Najoung Kim**[1,†]
[1]Boston University, [2]University of Amsterdam, [3]Umeå University,
[4]Vrije Universiteit Amsterdam, [5]TTIC, [6]The University of Texas at Austin

https://taxonomigqa.github.io

## Abstract

Does vision-and-language (VL) training change the linguistic representations of language models in meaningful ways? Most results in the literature have shown inconsistent or marginal differences, both behaviorally and representationally. In this work, we start from the hypothesis that the domain in which VL training could have a significant effect is lexical-conceptual knowledge, in particular its taxonomic organization. Through comparing *minimal pairs* of text-only LMs and their VL-trained counterparts, we first show that the VL models often outperform their text-only counterparts on a text-only question-answering task that requires taxonomic understanding of concepts mentioned in the questions. Using an array of targeted behavioral and representational analyses, we show that the LMs and VLMs do not differ significantly in terms of their taxonomic knowledge itself, but they differ in how they represent questions that contain concepts in a taxonomic relation vs. a non-taxonomic relation. This implies that the taxonomic knowledge itself does not change substantially through additional VL training, but VL training does improve the *deployment* of this knowledge in the context of a specific task, even when the presentation of the task is purely linguistic.

## 1 Introduction

Humans readily integrate perceptual and linguistic signals to form generalizable mappings from semantic information to language, allowing them to reason about concepts beyond their immediate environment [57, 18]. Approaches to concept grounding in AI, which traditionally relied on annotated datasets to specify how language links to people, objects, and events [72, 26], have rapidly shifted in light of the impressive capabilities of vision-language models (VLMs).

Many standard VLMs [33, 30, i.a.] often build on top of a pretrained language model (LM) by adding visual conditioning to its next token prediction task, often also updating the parameters of the language model. Analyses of VLM capabilities often focus on the multimodal tasks this additional modality enables. But (how) does this vision-and-language (VL) training change the linguistic capacity of the model? Answering this question requires comparing VL-tuned LMs to their original LM counterparts. Empirical evidence in this literature is rather sparse, often comparing such "VLM-LM minimal pairs" on general benchmarks such as MMLU [17] and GLUE [65]. In this paper, we consider a more targeted investigation (like [73]) of VLM-LM pairs in a particular domain:

---

*,†: Equal contribution; Code can be found at https://github.com/tinlaboratory/taxonomigqa

lexical-conceptual knowledge, specifically its taxonomic organization (e.g., *a cat is an **animal***). Evaluation of taxonomic knowledge has been of continued interest within the Natural Language Processing [14, 32, 43, 45] and Computer Vision communities [2, 62, 48]—however, to the best of our knowledge no work so far has compared *minimally* differing VLM-LM pairs in terms of how well they can reason taxonomically.

To this end, we develop `TaxonomiGQA`, a synthetically augmented *text-only* version of the popular visual-question answering (VQA) dataset GQA [19], where a subset of WordNet [40] hierarchy is used to create questions that require taxonomic knowledge. On comparing 7 widely used VLM-LM minimal pairs, we find most VLMs to consistently outperform their LM counterparts, despite the fact that the QA task is text only. We put forth two hypotheses to explain these results. **H1:** VL training fundamentally alters the (task-agnostic) taxonomic knowledge in LMs; and **H2:** VL training improves the ability of the LM to *deploy* its (largely unchanged) taxonomic knowledge in tasks that require its usage. Through a series of controlled behavioral and representational analyses, we find evidence that supports H2 relative to H1. Finally, we conduct a preliminary investigation where we relate the successes of VLMs over LMs to the visual similarities between the hyponym-hypernym categories we have tested in our work. Here we find initial evidence that suggests that VLMs especially perform well at answering questions about hyponym-hypernym pairs that are visually similar, leaving open areas of interesting future research for a more precise characterization of the role of visual input.

## 2 Related Work

**Influence of vision on language in VLMs** There are two main strands of empirical work measuring the influence of the additional visual modality on models' *linguistic* behavior and representations. The first line of work compares VLM and LM performance on downstream text-only benchmarks. The results are mixed: for instance, FLAVA [60] noted around 8% point gains over the base masked language model on GLUE-style NLP tasks (although the evaluation setting involved finetuning). On the other hand, Molmo has been reported to be outperformed by its base LM, Qwen, on text-only benchmarks like MMLU [8]. Generally, more evidence exists in favor of multimodal training hurting text-only task performance [21, 37] and this observation has been used to argue for freezing the language part of the model during multimodal training [12]. The second line of work conducts more targeted comparisons of VLMs and LMs, examining whether additional vision training leads to differences in representations of syntactic categories [66] and performance on tasks that require more "grounding" [73], but the findings overall have indicated no substantial differences. We contribute to this line of work by showcasing a context where there is a non-trivial difference brought about as a result of VL training. In particular, we show that while VL training does not fundamentally alter task-agnostic representations of taxonomic knowledge in LMs (in line with prior work), it does improve the *deployment* of this knowledge in the context of a question-answering task.

**Taxonomic knowledge and its deployment** Taxonomic knowledge has long been a topic of interest in cognitive psychology [35, 46], and has also often been used to analyze conceptual organization in LMs [14, 32, 45, 44]. Work that tests its functional consequences, such as property inheritance [43, 58, 56] and inductive generalization [42, 13], found strong evidence that while LMs do learn explicit taxonomic knowledge, they struggle to deploy it in taxonomically sensitive tasks [43]. Taxonomic knowledge has also been evaluated and analyzed in multimodal models. For instance, Pach et al. [48] show that the internal structure of neurons in models such as CLIP are often in alignment with existing taxonomies. Our work contributes to this line of work by proposing a level-ground comparison between minimally differing LMs and VLMs, narrowing in on the precise ways in which additional VL training may or may not alter the nature of this knowledge.

## 3 Behavioral testing of minimal pair VLMs and LMs with `TaxonomiGQA`

The question we are interested in answering concerns the *change* that VL training introduces to the lexical-conceptual knowledge of a model. This requires a shared evaluation that can be applied to both VLMs and LMs. We discuss below how we designed this evaluation as well as our findings about a range of VLM-LM pairs from this evaluation.

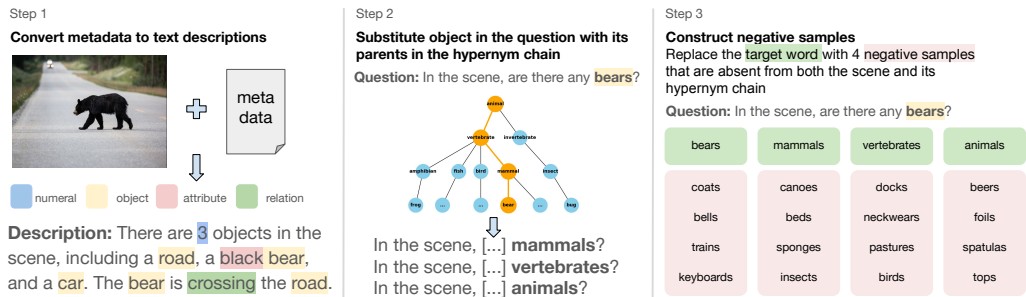

Figure 1: The three-step pipeline to create `TaxonomiGQA`.

## 3.1 Dataset design

We created a QA dataset that requires taxonomic understanding based on GQA [19], named `TaxonomiGQA`. A datapoint in GQA consists of an image of a scene, a question about this scene, and metadata that includes a scene graph of the objects, their attributes, and relations between the objects. We applied a three-step modification (each step illustrated in Figure 1) to create a *text-only* dataset, since our goal is to systematically compare the LM and VLMs' taxonomic competence. (1) Convert the scene graph into a purely textual description of the scene programmatically using hand-crafted templates; (2) For each question that contains a word that corresponds to a node in our reference taxonomy, substitute the word to its hypernym; (3) For each substitution, create four negative samples following Misra et al. [43], by substituting the target word with a word that is not in its hypernymy chain. Below we describe the resources, filtering, and sampling details used to create `TaxonomiGQA`.

**Taxonomy** To construct the reference taxonomy, we first extracted all unique noun lemmas ($N = 1216$) that appeared in the GQA questions, and annotated their senses in the WordNet taxonomy, these serve as the leaf nodes of our taxonomy. Next, for each noun, we extracted its hypernym chain (e.g., dog < canine < mammal < vertebrate < animal) from WordNet, rejecting hypernyms that were too abstract (determined manually), e.g., *entity, material, conveyance*. 315 concepts were removed as a result, many of which often had abstract entities in their hypernym chains or had non-ideal WordNet categorization (e.g., *bubble* as a member of *ball*), leaving us with 901 unique chains. (See Appendix C for more details.)

**Dataset construction** We applied a multi-stage filtering process to the validation split of GQA (10,696 images/scenes and 488,293 questions) to obtain our base questions. We first applied scene-level filtering by excluding scenes containing more than 20 annotated objects or any repeated object labels to avoid ambiguity in referring expressions in text. For each remaining scene, we applied question-level filtering to retain questions that refer to a single object (excluding any that mention multiple objects) and whose hypernyms do not overlap with those of any other object in the scene. Next, we balanced the dataset by randomly sampling 40 questions per scene in proportion to each scene's question type ratios. We further filtered the questions by answer type and restricted the dataset to yes/no questions to facilitate the substitution step. This reduced our taxonomy to 314 unique chains. In the base questions remaining after filtering, we substituted each target concept with each of its hypernyms in its hypernym chain to obtain the substituted questions. Then, we created negative sample questions by substituting the target concepts with concepts that are not in their hypernym chain, discarding question types where this substitution was not possible due to the introduction of presupposition failure (e.g., questions such as *Is the color of the dog brown?* when there is no dog in the image). This ended up eliminating more hypernym chains (which were only present in the discarded question types), leaving us with 126 final chains. More details about this negative sampling pipeline is given in Appendix D.

**Dataset statistics** The final dataset contains 1,342 unique images/scenes, 29,604 positive sample instances (9,334 targeting leaf node concepts, 20,270 targeting hypernym-substitutions), and 4 negative samples for each positive sample, amounting to 148,020 total instances. There are 276 hyponym-hypernym pairs, 126 unique hypernym chains, 88 unique hypernyms, and 24 top-level categories (e.g., *animal, vehicle,* etc.).

## 3.2 Metrics

We propose metrics designed to be sensitive to taxonomic structure (cf. [62]). The design principles are: (1) be sensitive to hierarchical relationships between two concepts; (2) anchor expectations on taxonomic knowledge conditioned on the model's success at foundational or prerequisite tasks; and (3) provide insight into robustness, including contrasting the performance on both positive and negative samples. By grounding our metrics in these properties, we move beyond correctness and toward a more systematic assessment of model performance.

As preliminaries, each instance, $X_i = (q, q^n_{1...4})$ in `TaxonomiGQA` consists of a positive sample question, $q$, about some leaf-level category (target concept), coupled with a set of 4 negative sample questions, $\{q^n_{1...4}\}$, where the target concept in the original question is now replaced by a negative-sample concept, as described in Section 3. Next, for each instance, we have a set of $k_i$ hypernym-substituted instances, $\{X^{s,i}_1, \ldots, X^s_{k_i}\}$, where each item $X^s_j$ is an instance but with the original category *substituted* with a category in its hypernym chain, along with their own 4 negative samples. Finally, we use a function, $\texttt{correct(.)} \to \{0, 1\}$, which accepts an instance $X$, and returns 1 iff. the model correctly answers the positive sample question *and* all four negative sample questions, and 0 otherwise. Using these preliminaries, we propose the following metrics:

**Overall Accuracy**  measures the proportion of time the model correctly answers all original, unsubstituted, and hypernym-substituted instances, treating each instance as separate item.

$$\text{Overall} = \frac{1}{N + \sum_{i=1}^N k_i} \left[ \sum_{i=1}^N \left( \texttt{correct}(X_i) + \sum_{j=1}^{k_i} \texttt{correct}(X^s_j) \right) \right] \tag{1}$$

**Conditional Accuracy**  measures the proportion of time the model correctly answers hypernym-substituted instances, conditioned on the fact that the model correctly answered the original, unsubstituted instance correctly. That is, if there are $N_{sel}$ original instances that the model answered correctly, the metric is calculated by:

$$\text{Conditional} = \frac{1}{\sum_{i=1}^{N_{sel}} k_i} \sum_{i=1}^{N_{sel}} \sum_{j=1}^{k_i} \texttt{correct}(X^s_j) \tag{2}$$

**Hierarchical Consistency**  proposed by Wu et al. [68, originally named "Hierarchical Consistence Accuracy"] measures a stricter form of accuracy relative to the previous ones, as the proportion of time the model correctly answers the original unsubstituted instance *and* all of its corresponding hypernym-substituted instances. Using our notation, this is measured as:

$$\text{HC} = \frac{1}{N} \sum_{i=1}^N \texttt{correct}(X_i) \prod_{j=1}^{k_i} \texttt{correct}(X^s_j) \tag{3}$$

All of the metrics incorporate robustness to negative samples (using the `correct()` function). Conditional Accuracy is stricter than Overall, ruling out cases where the model succeeds at higher level categories without correctly answering questions about the target object. HC requires the model to answer all questions about a hypernym chain correctly, being the strictest measure. That is, if the model fails to answer questions about *canines* then all *dog/wolf/fox* questions will be penalized. This is the most faithful to the chain in the reference taxonomy but may be considered overly strict.

## 3.3 Models

We selected seven LM-VLM model pairs, where the LM has been reported to be the base model that the VLM has been trained on top of, following the approach of [24]. The selected pairs are: (1) **Llama-3.1-8B** vs. **MLlama-3.2-11B** [12]; (2) their **instruct versions**; (3) **Vicuna** vs. **Llava-1.5-7B** [33]; (4) **Mistral-v0.2-Instruct** [22] vs. **Llava-Next** [34]; (5) **Qwen2-7B** [70] and **Molmo-7B-D** [8]; (6) **Qwen2-7B-Instruct** vs. **Llava-OneVision** [29]; and (7) **Qwen2.5-7B-Instruct** [71] vs. **Qwen2.5-7B-VL-Instruct** [4]. See Appendix A for more details. Since `TaxonomiGQA` consists of Yes/No questions, we sampled from a constrained probability distribution of Yes and No tokens from the models' output vocabulary, allowing for surface form variation such as casing and space-prefixing.

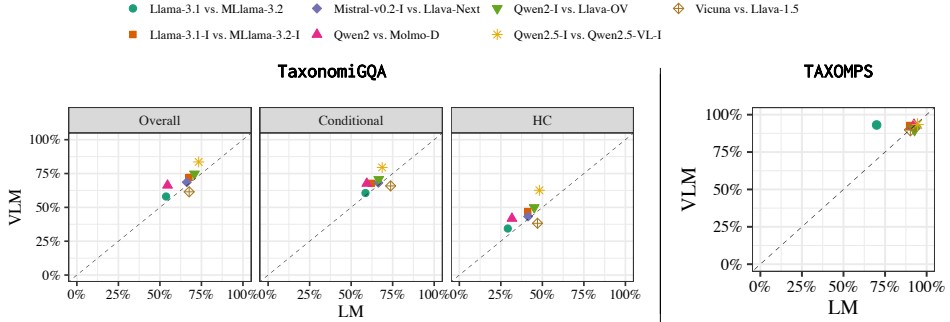

Figure 2: Performance of VLM-LM model pairs on `TaxonomiGQA` (Section 3) and `TAXOMPS` (Section 4.1). Points above the line indicate that VLM outperforms LM.

## 3.4 Results

The results are shown in Figure 2: points above the diagonal denote model pairs where the VLM outperforms the LM counterpart, and points below denote model pairs where the LM outperforms the VLM. We observe a consistent trend (with a single exception of Vicuna vs. Llava-1.5) where the VLMs outperform their LM counterpart, even though the presentation of the task was purely linguistic.[1] We rule out the possibility that the VLMs are performing better due to having been trained on GQA directly by running a control experiment where we give only the question (without the scene description) to the VLMs—if VLMs have encountered the original GQA instances during training, they may have learned question-label associations used in our dataset. Our results (Appendix E) show that most VLMs do not perform substantially above chance. Since the text descriptions are newly introduced in `TaxonomiGQA`, we can safely rule out the hypothesis that VLMs' improvements are due to having been trained on GQA. Accepting the trend of VLMs outperforming LMs on `TaxonomiGQA` as a genuine improvement, we conduct analyses that aim to explain this result more in the subsequent sections. When we are not analyzing all model pairs, we focus on Qwen 2.5-Instruct vs. Qwen 2.5-VL-Instruct, since the performance gap between VLM and LM was the most salient with this pair, especially on stricter metrics (conditional accuracy and HC).

**Generalizability of our finding** We conducted a supplementary experiment using the dataset from Rodriguez et al. [56]—another case where task contexts require the deployment of taxonomic knowledge—to verify the generalizability of our finding outside of `TaxonomiGQA`. Here, LMs were evaluated on their projection of novel properties (e.g., *is daxable, has feps*, etc.) from hypernyms to their hyponyms. Results on this dataset (shown in Table 5 in Appendix F) are qualitatively in line with those on `TaxonomiGQA`: VLMs were substantially better than their LM counterparts in 5 out of 7 VLM-LM pairs. Furthermore, concurrent work by Tan et al. [63] that also investigates taxonomic understanding in minimally differing LMs and VLMs corroborates our finding. In their results, VLMs (with the exception of LlaVa-OneVision, and to a milder extent, InternVL-8B) showed improved performance on 4 out of 5 taxonomic understanding benchmarks.

## 4 H1: VLMs' taxonomic knowledge aligns better with reference taxonomy

One possible reason that VLMs are performing better on `TaxonomiGQA` could be due to the difference in their underlying (task-agnostic) taxonomic knowledge, and in particular, in a way that better aligns with the reference taxonomy used to create the hypernym-substituted questions. We test this hypothesis about taxonomic knowledge difference in three different ways: (1) through a QA task that directly elicits taxonomic judgments; (2) through an analysis of the hierarchical organization of concepts in the models' representation space; and (3) through similarity analysis on the embeddings.

---

[1]For the curious: see Appendix E for how the VL models perform on our text-only QA vs. VQA.

## 4.1 Directly eliciting taxonomic judgments through Taxonomic Minimal Pairs (`TAXOMPS`)

Since `TaxonomiGQA` *presupposes* taxonomic knowledge rather than eliciting it directly, we first checked whether VLMs and LMs differed in their ability to directly answer questions about taxonomic relations. To this end, we introduce `TAXOMPS` (Taxonomic Minimal Pairs), a dataset which consists of questions of the form "*Is it true that a $C_1$ is a $C_2$?*" where $C_1$ (*cat*) and $C_2$ (*feline*) are concepts that are in a hypernymy relation, and negative samples where $C_2$ is replaced by a concept that is not the hypernym of $C_1$ (*vehicle*), following Misra et al. [43]. We constructed `TAXOMPS` directly from the final taxonomy used in our `TaxonomiGQA` analysis—i.e., 276 total hyponym-hypernym pairs, each coupled with 4 negative samples (same as in `TaxonomiGQA`), yielding 1380 questions. We use Overall Accuracy as our performance measure (since there is no conditional analog), following Section 3. That is, an instance is considered correct iff. the model answers questions with the hyponym-hypernym pairs (*Is it true that a **cat** is an **animal**?*) with a Yes while answering No to the negative sample questions (*Is it true that a **cat** is a **vehicle/fruit/tool/vegetable**?*).

Figure 2 shows our results. With the exception of Llama-3.1 vs. MLlama 3.2, most VLM-LM pairs perform quite similarly (and well) on `TAXOMPS`. This suggests that additional VL training does not in general alter the basic taxonomic membership judgments of a language model.

## 4.2 Lexical representations of taxonomic knowledge

Can the alignment with reference taxonomy be observed representationally, although not by direct elicitation? We tested whether the lexical representations in VLMs align better with the reference taxonomy than LMs via their hierarchical organization and hypernym-hyponym embedding similarity.

### 4.2.1 Hierarchical taxonomic structure

Park et al. [52] propose a method to analyze the latent hierarchical taxonomic structure of an LM, based on ideas including the linear representational hypothesis [39, 53] and causal separability of concepts [67], finding that taxonomic hierarchies (*dog < canine < mammal*...) are encoded as orthogonalities in LMs' transformed unembeddings. Therefore, one way we may observe the effect of VL training on the taxonomic knowledge of the LM is via differences in this hierarchical structure.

We applied Park et al. [52]'s method to transform the unembedding space in our models to a space where the inner product between two concepts' vectors is sensitive to the hierarchical relation between them. Then, we compared VLM-LM pairs in terms of their pairwise cosine similarities between concepts in their unembeddings' causal inner product space (as established in [53]). In addition, we used the large WordNet hierarchy (a superset of our taxonomy) originally used by [52] to compare the pairwise similarities of concepts in VLM and LM to that of the pairwise path-similarities between concepts in WordNet. We conducted these comparisons using Representational Similarity Analysis [25], which computes the Spearman's correlation between two matrices' (flattened) upper triangular matrices, treating it as the representational similarity between the two spaces. We conducted RSA between three representational spaces: VLM, LM, WordNet. Greater RSA value between two spaces indicates greater similarity between. To account for potential variance, we sampled 100 subsets (of size $100 \times 100$ each) from the full pairwise matrices and report the mean and standard deviation of the RSA correlations across all subsets.

This analysis (Table 1, left) shows that the hierarchical organization of concepts (as defined by [52]) is mostly shared between the VLM and LM, indicated by the consistently similar RSA scores when comparing VLMs and LMs to WordNet, as well as the high similarity between the VLM and LM when directly compared (all RSA scores $\geq 0.95$). Interestingly, the Qwen 2.5 and Molmo pairs, the two model pairs that showed the most salient advantage of VLMs in Figure 2 had the lowest VLM-LM RSA scores: 0.95 and 0.96, respectively. However these values are still very high in terms of raw correlation, suggesting that they are still fundamentally similar. The pairwise similarities for VLMs, LMs, and WordNet can be visually inspected in Figure 5 in Appendix G.

### 4.2.2 Embedding similarities of taxonomic relations

Another way in which taxonomic relations can be investigated is via vector similarity—we tested whether the lexical embeddings (i.e., uncontextualized representations) corresponding to concepts in our reference taxonomy are more similar to embeddings of their hyponyms, relative to embeddings

Table 1: **Left:** RSA comparisons of hierarchically sensitive pairwise similarities [52] in the unembedding spaces of VLM-LM pairs, and pairwise path-similarities from the WordNet (WN) Noun Hierarchy. Subscripts show standard deviation (hidden if under 0.01). **Right:** Differences ($\Delta$) in cosine similarities between positive concept pairs (i.e., in a hypernymy relationship) and negative samples from the taxonomy in `TaxonomiGQA`, computed using VLM and LM static-embedding layers.

| Minimal Pairs | RSA using Park et al. [52] | | | Raw Embeddings | | | |
|---|---|---|---|---|---|---|---|
| | (VLM, WN) | (LM, WN) | (VLM, LM) | $\Delta_{\text{VLM}}$ | $\Delta_{\text{LM}}$ | $t$ | $p$ |
| Vicuna vs. Llava-1.5 | $0.43_{\pm 0.04}$ | $0.43_{\pm 0.04}$ | 0.99 | 0.02 | 0.02 | 1.09 | 0.27 |
| Mistral-v0.2-I vs. Llava-Next | $0.42_{\pm 0.04}$ | $0.42_{\pm 0.03}$ | 0.99 | 0.04 | 0.04 | 1.19 | 0.23 |
| Qwen2.5-I vs. Qwen2.5-VL-I | $0.38_{\pm 0.05}$ | $0.39_{\pm 0.04}$ | **0.95** | **0.03** | **0.04** | -7.51 | <.001 |
| Llama-3.1 vs. MLlama-3.2 | $0.40_{\pm 0.04}$ | $0.41_{\pm 0.04}$ | 0.99 | 0.04 | 0.04 | 1.34 | 0.18 |
| Qwen2-I vs. Llava-OV | $0.40_{\pm 0.04}$ | $0.40_{\pm 0.05}$ | 0.99 | 0.06 | 0.06 | 0.82 | 0.41 |
| Qwen2 vs. Molmo-D | $0.38_{\pm 0.04}$ | $0.39_{\pm 0.04}$ | **0.96** | 0.05 | 0.05 | - | - |
| Llama-3.1-I vs. MLlama-3.2-I | $0.40_{\pm 0.04}$ | $0.40_{\pm 0.04}$ | 0.99 | 0.04 | 0.04 | -0.09 | 0.92 |

of non-hyponyms in our taxonomy. We computed the similarity between each target concept and its hyponym, as well as between the target concept and four randomly sampled non-hyponym concepts (same as in Section 3.1). Then, we computed the difference between target-hyponym similarity and the average similarity between the target and the negative samples. We tested whether this difference is greater in VLMs than LMs, which would mean that VLM embeddings encode hypernym-hyponym relations more similarly than non-hypernym-hyponym relations. This hypothesis is *not* borne out: Table 1 (right) shows that this holds for no VLM-LM pairs (the significant effect in Qwen2.5-I vs. Qwen2.5-VL-I is in the opposite direction).

# 5 H2: VLMs are better at deploying taxonomic knowledge

As mentioned in Section 4.1, solving a downstream task presupposes the domain knowledge recruited, and requires this knowledge to be correctly deployed in the context of the specific task. Hence, solving `TaxonomiGQA` requires (1) taxonomic knowledge and (2) its deployment specifically for scene description-based QA. Our analyses in the previous section did not show convincing evidence in support of the hypothesis that the underlying taxonomic knowledge differs substantially in our VLM-LM pairs. In light of this mostly negative result, we turn to our second hypothesis: VLMs are better at *deployment* of taxonomic knowledge. To test whether there is a difference when taxonomic knowledge is incorporated into the specific task context, we used *contextual* similarity of lexical representations and a Principal Component Analysis (PCA) of representations of questions. These analyses let us examine both the contextualized lexical representations as well as the holistic representation of the full question context. We used the Qwen2.5 pair in both analyses.

**Data** To control for the confounding effect of the target label (Yes/No) when analyzing contextualized representations, we used a subset of `TaxonomiGQA` that has the same ground truth label for both positive and negative samples. In our dataset, this only includes cases where the ground truth answer is No. We further filtered this dataset to instances where the models got the original, unsubstituted question right, and used the models' Conditional Accuracy on substituted questions as the target of study. This gave us us 37,790 and 40,145 samples for Qwen2.5-I and Qwen2.5-VL-I, respectively.

## 5.1 Contextualized representation similarity

Our first analysis aims to relate the behavioral outcome of a model for each question to the representational structure of the concepts in context. To this end, we investigated the contextualized representations of a target concept in the scene description in terms of their similarity to representations of its hypernym in the question (e.g., *There is a $\textbf{dog}_{hypo}$ on a yellow surfing board [...]. In the scene, are there any $\textbf{mammals}_{hyper}$?*). The quantitative hypothesis is that greater contextualized hypernym-hyponym similarity (e.g., sim(**dog**, **mammal**) compared to hyponym similarity with negative samples (e.g., sim(**dog**, **fruit**) from *There is a $\textbf{dog}_{hypo}$ [...]. In the scene, are there any $\textbf{fruits}_{neg}$?*) would predict how well the model can answer the `TaxonomiGQA` questions. We used the 4 negative samples from `TaxonomiGQA`, and then fit a logistic regression model to predict model

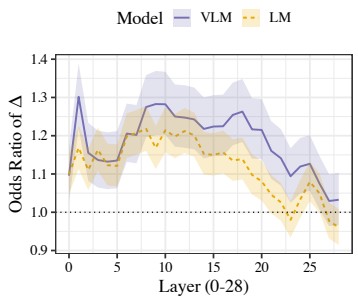

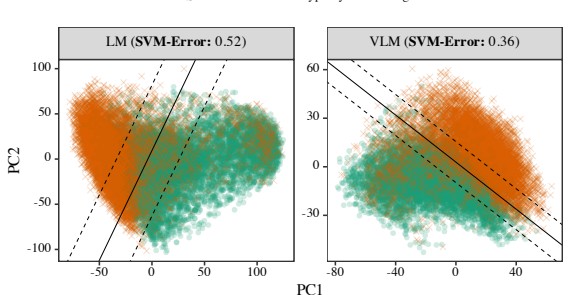

(a) Odds ratio of the difference between co-sine similarity of hyponym and hypernym, and max cosine similarity of hyponym and negative samples ($\Delta$) in estimating model correctness, across layers.

(b) PCA projections of the last hidden state representations of questions containing hypernym (●) vs. negative sample substitutions (×), extracted from Qwen2.5-I (LM) and Qwen2.5-VL-I (VLM), along with separation lines fitted using an SVM classifier. SVM Error denotes the margin error *and* the classification error.

Figure 3: Contextualized representational analysis on Qwen2.5-I and Qwen2.5-VL-I.

correctness (measured using the `correct()` function from Section 3.2) using the difference in cosine similarity between hypernym-hyponym, and maximum[2] cosine similarity of 4 hyponym-negative sample pairs. Here, an odds ratio (of the difference term) being greater than 1 indicates that the similarity of the hyponym to hypernym (relative to negative samples) is more strongly associated with the model correctly answering questions, while the opposite is true if the odds ratio is less than 1. We performed this analysis using representations from every layer in the Qwen2.5 model pair, and took the maximum similarity in cases where the hyponym is mentioned in the scene more than once.

Figure 3a shows the layerwise odds ratios of the difference in similarities between concept pairs (sim-$\Delta$; discussed above), in predicting model correctness, for both models. For most layers, we see odds ratios greater than 1.0, indicating a positive association between sim-$\Delta$ and model correctness for both model classes. At the same time, the VLM odds ratios are often greater than those of the LM, with the LM odds ratios sometimes even veering off below the 1.0 level (which would suggest an association of sim-$\Delta$ with *wrongness* as opposed to correctness. Overall, this suggests that VL training helps establish stronger connections between model representations and behavior in task contexts requiring deployment of taxonomic knowledge.

## 5.2 Principal Component Analysis (PCA) of question representations

Like in the previous analysis, we focused on the distinguishability of hypernym-hyponym relations from non-hypernym-hyponym relations, but considered whether this is captured in the representation of the question context from data used in the previous section. Following Alhamoud et al. [1] (who tested negation sensitivity in VLMs), we took the last hidden state of the final layer of the text decoder to be the summary representation of the full context. Then we asked whether representations of questions that contain a hypernym-hyponym relation (e.g., *dog-canine*) are separated from representations of questions that contain a non-hypernym-hyponym relation (e.g., *dog-bird*) via PCA.

Figure 3b shows the first two principal components (PCs) of the question representations from the VLM & LM, with hypernym (green) vs. negative sample substitutions (orange) color coded. We see that the two types are largely visually distinct in both models, suggesting that their question representations do encode differences in terms of the taxonomic relations tested. To quantify (linear) separability, we fit a soft-margin support vector machine (SVM) classifier [7] on the first two principal components of the representations extracted from each model separately, and measured its error on the PC-representations—greater the error, the poorer the separability. We found that the SVM error of the PCs of VLM representations is substantially lower than that of the LM, demonstrating that taxonomic distinctions are more linearly separable in VLM question representations. This complements the results from the previous analysis of contextualized embeddings, and suggests genuine differences in the representational states of the VLM and LM when the task contexts require taxonomic reasoning.

---

[2]We note that taking the average instead of maximum results in substantially weaker trends.

## 5.3 On the distinction between knowledge possession and deployment

Collectively, our results highlight that VL training does not change the underlying taxonomic knowledge within LMs, but rather affects its *deployment* in task contexts that require sensitivity to taxonomic knowledge. We see two specific reasons why this distinction could be important.

First, storing or representing knowledge differs from learning its "functional consequences" [46]. A model may robustly encode category information (e.g., that *robins are birds*), yet fail to recruit this knowledge when the context demands it. `TaxonomiGQA` is designed precisely to probe this aspect of deployment: in order to be successful, a model must not only *store* taxonomic knowledge, but *use* it appropriately when answering questions. Practically, teasing apart knowledge possession and deployment can inform decisions about (post-)training data selection: if a model's limitations stem from representational gaps, additional encyclopedic knowledge may help; if the issue lies in deployment, more diverse contextual supervision, involving contexts in which such knowledge is recruited, could be more effective. This distinction can also motivate solutions; for instance transferring task vectors from models that are better at deployment [16].

Second, the distinction has implications for cognitive and philosophical interpretations of multimodal learning—in particular, for drawing appropriate conclusions about the roles of linguistic versus (added) extralinguistic exposure. For instance, the platonic representation hypothesis [20] suggests that models trained on sufficiently large amounts of data converge toward similar internal representations, irrespective of modality. Our findings provide a complementary perspective to this hypothesis. While independent unimodal models might converge to similar taxonomic representations, combining information from multiple modalities can result in non-trivial changes that go beyond representational convergence (in our case, in terms of how knowledge is accessed and deployed).

## 6 *Why* might vision training help?

Our analyses so far have pinpointed *where* the meaningful behavioral and representational differences lie in the context of a taxonomic task when comparing a VLM-LM pair. However, we have not discussed *why* vision training would be beneficial. We present a preliminary investigation here, hypothesizing that visual similarity between members of concepts in a hypernym-hyponym relation is helpful information that VLMs can leverage. Some examples would be the visual similarity of members of *equine* and *horse* or *root vegetable* and *radish*. Of course, visual similarity will not be informative cues for *all* such relations, e.g., it would not be very helpful in better understanding the relation between *vertebrates* and its hyponyms, since there are few salient visual features shared by members of *vertebrate* (e.g., *fish, mammal, amphibian*...). This motivates a hypothesis that links visual information to model performance: high visual similarity between members of a hypernym and its hyponym would have a positive effect on model performance on questions probing that relation, but the effect would substantially vary depending on the target concepts.

**Method**  To test this hypothesis, we first estimated hypernym-hyponym visual similarities by computing the cosine similarity between the image representation of a leaf node object and the image representations of other objects within its parent node (i.e., its hypernym) for concepts in our taxonomy. The image representations are extracted from the target VLM's (Qwen2.5-VL-I) vision encoder. Importantly, the images themselves are sourced from an independent dataset (THINGS [15]) so that our conclusion is not tied to specific images in GQA. Rather, they are intended as estimates of visual similarity more broadly. More details about the image similarity computation is in Appendix H. Then, we tested the extent to which this similarity predicts Conditional Accuracy of the VLM on hypernym questions where it outperforms its LM counterpart, using a linear mixed-effects model. Specifically, we predicted Conditional Accuracy of the VLM between each hyponym-hypernym pair using the pair's visual similarity as a fixed effect, and included random slopes and intercepts for each hypernym (model formula: `cond_acc ~ viz_sim + (1 + viz_sim | hypernym)`).

**Results**  We found a significant global effect of visual similarity in predicting Conditional Accuracy ($b = 0.52, \mathrm{SE} = 0.19, p < .01$). These results are much weaker when using the text-only LM's Conditional Accuracy as the dependent variable ($b = 0.23, \mathrm{SE} = 0.17, p = 0.18$), suggesting that image similarity captures interesting properties related to the success of the VLM and uniquely so for VLMs. We also found interesting hypernym-specific random effects, where the effect of similarity

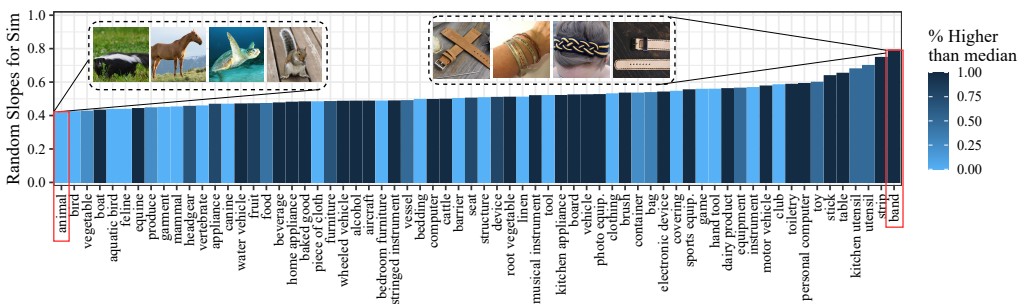

Figure 4: Hypernym-specific random effects of image similarity in predicting VLM accuracy on **TaxonomiGQA**. Greater values indicate closer association of visual similarity to model accuracy. Bar colors indicate percentage of hypernym-hyponym pairs that have above-median similarity.

varies greatly depending on the hypernym. Figure 4 shows the random slopes for each hypernym. This substantial individual variation aligns with our initial intuition that visual similarity would help some relations more than others. To quantify this intuition concretely, we annotated the higher level concepts in our taxonomy in terms of the % of the time the visual similarity to their hyponyms were above the median. This is meant to capture the difference between *equine* and *animal* we discussed earlier: members of *equine* are more similar to each other, more so than members of *animal* are (i.e., visually cohesive). The colors of the bars in Figure 4 are mapped to the degree of visual cohesion, where darker bars mean more cohesive. We see that the degree of cohesion generally lines up with effect sizes of similarity on predicting VLM performance, with mostly lighter bars on the left edge and darker bars on the right edge. The figure zooms into the concepts on either edge (left: *animal*, right: *band*), showing a sample of images corresponding to those concepts to illustrate the low visual cohesion of *animal* and high visual cohesion of *band*. Overall, the results present a promising lead into elucidating the source of improvement in VLMs, establishing a potential link between visual similarity, visual cohesion, and behavioral QA performance.

# 7 Conclusion

By building **TaxonomiGQA**, a text-only QA dataset that requires taxonomic understanding, we identified an interesting performance gap between VLM and their LM counterparts. That is, most VLMs consistently outperformed LMs under all metrics we adopted, despite this task being purely text based. We set out to pinpoint the source of this gain in VLMs. The first set of findings show that both behaviorally and representationally, there was no substantial difference between VLMs and LMs in their taxonomic knowledge, corroborating the general implications of [73, 66] that additional vision training does not fundamentally restructure the underlying knowledge. However, our second set of analyses show: (1) VLMs' contextual representation similarity of concepts in taxonomic relation in higher layers better predict success on **TaxonomiGQA**, and (2) VLM representations of questions containing taxonomic relations and questions that do not are better linearly separable, suggesting that VLMs have an advantage over LMs in adequately *deploying* taxonomic knowledge. We furthermore conducted a preliminary investigation on *why* vision training helps, testing the hypothesis that visual similarity of members in the extension set of hypernym/hyponyms help VLMs learn more useful representations of these words for taxonomic tasks. The results showed that VLMs' behavioral success on **TaxonomiGQA** can be predicted by visual similarity between members of concepts in a taxonomic relation, and the prediction strength is modulated by the visually cohesion of the hypernym.

**Limitations and future work** Our analyses do not provide causal evidence for the relation between behavior on **TaxonomiGQA** and the analyzed representations. Gaining causal evidence would require analyses more closely tied to the training data and objective, which is challenging due to the scale of the models as well as the scarcity of open data models. Additionally, a caveat to our results is that VL-tuned models do encounter more text-data in addition to visual supervision. This confound can be teased apart in future work by training LMs on the text-only portion of the VL training data. Furthermore, our SVM-based separability analysis is only applicable to taxonomic distinctions that are linearly encoded, leaving room for future work to extend this to non-linear separability.

## Acknowledgments

We thank Yukyung Lee for her advice on creating better visualizations and Mahir Patel for earlier discussions on constructing the dataset, as well as for their general support. We also thank the anonymous NeurIPS reviewers and the Area Chair for helpful feedback. Pilot experiments for this work were conducted with the support of the PaliGemma Academic Program GCP Credit Award from Google awarded to the team. An in-person workshop partially dedicated to this work was funded by the Royal Netherlands Academy of Arts and Sciences (KNAW) Early Career Partnership. We acknowledge that the computational work reported on in this paper was performed on the Shared Computing Cluster which is administered by Boston University's Research Computing Services, as well as on the burrata machine at TTIC. Kanishka Misra is supported by the Donald D. Harrington Faculty Fellowship at UT Austin.

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

Table 2: Overview of model configurations used in our experiments, including size, modality, and model training details. The "Training Details" columns indicate whether the model was pretrained on GQA, trained with video data, or instruction-tuned. A checkmark (✓) denotes the presence of the corresponding training signal, (✗) indicates its absence, a bold question mark (**?**) represents unknown or unclear training status, and a blank cell indicates that the category is not applicable. Hugging Face identifiers are provided for reproducibility.

| Model | Size | Modality | Training Details | | | HF Identifier |
|---|---|---|---|---|---|---|
| | | | GQA Pretrained | Video Involved | Instruction Tuned | |
| Qwen2 | 7B | L | | | ✗ | Qwen/Qwen2-7B |
| Molmo-D | 7B | VL | ✗ | ✗ | ✓ | allenai/Molmo-7B-D-0924 |
| Llama-3.1 | 8B | L | | | ✗ | meta-llama/Llama-3.1-8B |
| MLlama-3.2 | 11B | VL | ? | ? | ✗ | meta-llama/Llama-3.2-11B-Vision |
| Vicuna | 7B | L | | | ✓ | lmsys/vicuna-7b-v1.5 |
| LLaVA-1.5 | 7B | VL | ✓ | ✗ | ✓ | llava-hf/llava-1.5-7b-hf |
| Qwen2-I | 7B | L | | | ✓ | Qwen/Qwen2-7B-Instruct |
| LLaVA-OV | 7B | VL | ✓ | ✓ | ✓ | llava-hf/llava-onevision-qwen2-7b-ov-hf |
| Mistral-v0.2-I | 7B | L | | | ✓ | mistralai/Mistral-7B-Instruct-v0.2 |
| LLaVA-Next | 7B | VL | ✓ | ✗ | ✓ | llava-hf/llava-v1.6-mistral-7b-hf |
| Llama-3.1-I | 8B | L | | | ✓ | meta-llama/Llama-3.1-8B-Instruct |
| MLlama-3.2-I | 11B | VL | ? | ? | ✓ | meta-llama/Llama-3.2-11B-Vision-Instruct |
| Qwen2.5-I | 7B | L | | | ✓ | Qwen/Qwen2.5-7B-Instruct |
| Qwen2.5-VL-I | 7B | VL | ✗ | ✓ | ✓ | Qwen/Qwen2.5-VL-7B-Instruct |

# A  Selected Model Pairs

Table 2 shows a list of model pairs used in this work, along with their metadata – parameters, modality, huggingface identifier, etc.

# B  Extended Related Work

**Multimodal Semantic Representations in Humans and Language Models.**  A central question in cognitive science and linguistics is how humans integrate perceptual and linguistic signals to form generalizable mappings from semantic and conceptual knowledge to language. Research exploring the cognitive and neural underpinnings of such knowledge supports the idea that language learning and processing is inherently multimodal, grounded in visual, motor, and affective experience [61, 64]. At the neural level, conceptual knowledge is proposed to be coordinated by a transmodal "semantic hub," allowing humans to flexibly attend to the modality that provides the most informative cue in context and to abstract over modality-specific input [55, 54]. Several NLP tasks now commonly employ multimodal representations [5], notably image captioning [59, 38] and visual commonsense reasoning [74, 31]. In embodied agents, linking physical actions to explicit linguistic representations has been shown to facilitate more effective concept learning [36, 6].

Computational representations can be optimized by identifying and exploiting semantic structure shared across modalities. Models trained on different modalities and objectives may converge on similar representations as they grow in size, forming a "platonic" structure shaped by statistical correlations across input that is modality agnostic [20]. Unified representations and architectures have been argued to better support multimodal processing and reasoning by reshaping how models reference and access perceptual and linguistic features, both reflecting the "semantic hub" structure found in humans and partially mitigating common biases found in unimodal models [60, 10, 69]. These methods can enable the implicit grounding of language in perceptual features such as spatial awareness and sound, even in text-only models [47].

For vision and language modalities, the Visual Question Answering (VQA) task [3] has inspired work on joint language and image understanding using on compositionality, consistency metrics, and knowledge-enriched prompting [23, 11, 9]. Focused benchmarks like VALSE [51], which tests

models' ability to ground linguistic phenomena in the visual modality, and interpretability methods such as MM-SHAP [49] and CC-SHAP [50], measure how VLMs integrate and rely on visual versus textual information. Findings show that VLMs often underuse visual input for reasoning, yet rely on it more heavily for generating explanations, especially in chain-of-thought (CoT) settings. These findings highlight that contributions of each modality in VLMs are uneven and task-dependent, challenging assumptions of uniform multimodal integration. An open question thus remains as to whether multimodal training indeed changes conceptual content, or instead how that content is accessed and applied. Our research explores this in a unique setting where VLM/LM minimal pairs share the textual component.

## C   Taxonomy Filtering and Annotation

Before manual annotation and removal of specific chains, we first identified 52 highly abstract concepts[3] (e.g. *entity, conveyance, act*, etc.) to be removed from all chains. After generating the initial hypernym chains, we conducted a second round of manual inspection to identify and address cases of "non-ideal" categorizations, defined as instances where either (a) the assigned hypernym was not the canonical category of the object (e.g., *bubble* categorized as *circle*), or (b) the chain consisted solely of abstract elements that were missed during the first filtering step. Through this process, we identified 611 problematic cases. Of these, 296 were corrected by querying WordNet for alternative hypernym chains, leaving 315 unfixable cases; these were subsequently removed (as reported in the main text).

## D   Negative Sampling Details

Table 3: Nine question types in **TaxonomiGQA**. Each question type is illustrated with an example and the total number of instances of that category. Question types ending in "C" have "no" as the correct answer ("C" stands for counterfactual); all others have "yes" as the correct answer, consistent with the design of GQA.

| Question Type | Sample Question | Counts |
|---|---|---|
| exist | Are there any dogs? | 29030 |
| existAttr | Are there any boats that are white? | 16405 |
| existAttrNot | Are there dogs in this scene that are not white? | 15300 |
| existAttrC | Do you see dogs that are white? | 16010 |
| existAttrNotC | Do you see a fork that is not silver? | 16440 |
| existThat | Are there any tables in the picture that are wooden? | 20435 |
| existThatNot | Is there a television in the image that is not off? | 4120 |
| existThatC | Is there a boat that is green? | 19985 |
| existThatNotC | Is there a watch in the image that is not on? | 3670 |
| existMaterial | Do you see a fence that is made of wood? | 1750 |
| existMaterialNot | Is there a bench that is not made of wood? | 1465 |
| existMaterialC | Are there any lace tablecloths? | 1650 |
| existMaterialNotC | Are there forks that are not made of metal? | 1760 |

After dataset filtering, we identified 32 types of questions. 19 of them were excluded because substituting the object in the question with one not present in the scene could result in presupposition failures. For the remaining types, we sampled four negative objects for each question based on the following three criteria: the sampled argument is (1) not present in the scene, (2) not in the original argument's hypernym chain, and (3) associated with the same set of attributes as the original arguments defined in GQA metadata. Due to inconsistencies and errors in the GQA metadata, we manually[4] verified the attribute matches to ensure the naturalness and validity of each substitution.

---

[3]These concepts are listed in the `GQA Hierarchies First Pass.ipynb` notebook located under `notebooks/` in our repository.

[4]We remove attributes that introduce non-standard property attribution, such as "happy trees", "swimming flowers", "fluffy apple".

This process resulted in a final dataset consisting of 13 question types and reduced our taxonomy to 126 unique chains. Details of question types, examples, and statistics can be found in Table 3.

# E    VQA vs. Text and a Question-only Control

Table 4 shows results from evaluating VLMs on the original GQA questions across different formats: (1) the original VQA setup, conditioned on an image; (2) the Text setup, where they are conditioned on the scene description; and (3) a Question-only control where we condition them only on the question, without any context.

While it is difficult to compare between the VQA and the text setup, we see stark differences in the absolute values of the accuracies. The VLMs seem to answer the (positive sample, unsubstituted) questions with very high accuracy (sometimes near-perfect) relative to their performance on the subset of the VQA task we have used in this work. Next, the VLMs are substantially worse at the question-only baseline than they are in the text version, often times being closer to chance (50%). This question-only control is especially relevant for any potential concern readers might have about VQA data being present in the models' training set. Since models are largely worse off at these relative to the text version of the dataset, the potential presence of VQA in the model's training set is of little concern. One interesting observation here is that MLlama-3.2 (non-instruct tuned) performs similarly at the Question-only task and at the VQA task. This could suggest that it might not have been trained on VQA after all.

Table 4: Accuracies of VLMs on GQA questions when evaluated using standard VQA-based setup (i.e., with images) vs. Text (i.e., with scene descriptions), as well as a Question-only control (No image and no text). Evaluation data consists only the positive sample version of the dataset with unsubstituted questions taken verbatim from GQA [19]. Chance performance is 50%.

| Model | VQA | Text | Question-only |
|---|---|---|---|
| Molmo-D | 0.79 | 0.89 | 0.52 |
| MLlama-3.2-I | 0.79 | 0.92 | 0.58 |
| MLlama-3.2 | 0.63 | 0.91 | 0.60 |
| Llava-1.5 | 0.78 | 0.95 | 0.53 |
| Qwen2.5-VL-I | 0.81 | 0.98 | 0.49 |
| Llava-Next | 0.84 | 0.98 | 0.52 |
| Llava-OV | 0.87 | 0.99 | 0.60 |

# F    Supplementary Experiment on the Rodriguez et al. Dataset

Property inheritance testing datasets such as those introduced by Rodriguez et al. [56] involve attributing a novel, nonsense property (e.g., *is daxable*) to concepts given that their parents are attributed with it—e.g., *Given that birds are daxable, is it true that robins are daxable? Answer with Yes or No*. This particular dataset includes a robustness check in the form of a single negative sample per concept and includes 2016 positive samples, spanning 44 superordinate categories and 1281 subordinate categories.[5] We evaluated our model pairs on this dataset with minimal prompt tuning,[6] and found that, as shown in Table 5, all VLMs (except MLlama-3.2 and Llava-OV) outperformed their LM counterparts, reinforcing the robustness of the observed VLM > LM trend on models' behavioral performances on taxonomic understanding within a QA setting.

---

[5]This dataset has a larger set of categories than TaxonomiGQA, as its taxonomy is not constrained by visual scenes.

[6]We added the quantifier "all" to the premise - e.g., *Given that **all** birds are daxable, is it true that robins are daxable? Answer with Yes or No*, to make the logical inference more coherent. This modification improved the QA accuracy significantly, particularly for Qwen2.5-VL-I and MLlama-3.2.

Table 5: Performance comparison between LMs and VLMs on the Rodriguez et al. dataset. $\Delta$(VLM–LM) denotes the difference in accuracy between the VLM and its corresponding LM.

| Pair | LM | VLM | $\Delta$(VLM–LM) |
|---|---|---|---|
| Llama-3.1 vs. MLlama-3.2 | 0.50 | 0.68 | **0.18** |
| Llama-3.1-I vs. MLlama-3.2-I | 0.51 | 0.50 | -0.01 |
| Mistral-v0.2-I vs. Llava-Next | 0.69 | 0.78 | **0.09** |
| Qwen2 vs. Molmo-D | 0.60 | 0.81 | **0.21** |
| Qwen2-I vs. Llava-OV | 0.84 | 0.81 | -0.02 |
| Qwen2.5-I vs. Qwen2.5-VL-I | 0.78 | 0.83 | **0.05** |
| Vicuna vs. Llava-1.5 | 0.50 | 0.74 | **0.24** |

## G  RSA Heatmaps

We depict heatmaps showing the pairwise cosine similarities computed for the transformed Unembedding vectors of the Qwen2.5 model pair, as well as pairwise path-similarity from WordNet, in Figure 5. The LM and VLM matrices look quite similar, while the WordNet matrices are more sparse, showing clearer depiction of hierarhical structure. We computed similar plots for all other models but left them out due to large file sizes. Full plots can be viewed on https://github.com/tinlaboratory/taxonomigqa.

## H  More Details about Visual Similarity Analysis

To compute visual cosine similarity between two nodes–a leaf node object (e.g., *dog*) and one of its hypernyms (e.g., *vertebrate*)– we first needed a sufficient number of images for both. We used images from THINGS [15], a dataset with 26,107 high-quality, manually curated object-centric images of 1,854 diverse object concepts. Since the taxonomy in THINGS is more coarse-grained than ours, we aligned the taxonomies through the following steps: (1) Identify intermediate nodes that are missing in THINGS (e.g., *vertebrate*); (2) Collect leaf node objects present in THINGS and prompt a large language model (GPT-4o and Gemini 2.5 Pro) to identify which of them can be classified under the given intermediate node (e.g., *vertebrate*); 3) Manually verify the correctness of the selected objects. After aligning the taxonomies, we obtained visual representations for each node in our taxonomy from the Qwen 2.5VL-7B Instruct model. To do so, we modified the model's forward pass to extract hidden states immediately after the `merger.ln_q` RMSNorm layer within the `Qwen2_5_VLPatchMerger` module, and before the `merger.mlp` layer. These intermediate hidden states served as patch-level embeddings, which we mean-pooled to produce a 1280-dimensional representation for each image. We then computed cosine similarities between the visual representation of the leaf node (e.g., *dog*) and each of its hypernyms (e.g., *vertebrate*) by taking the mean of all image embeddings for the intermediate category node—similar to the prototype approach in [28]—and compute pairwise cosine similarities with each image from the leaf node.

## I  Experimental Resources

Dataset filtering for **TaxonomiGQA** was performed using multithreaded processing across 8 CPU cores and completed in approximately 3 hours. Negative sampling was carried out on a single CPU core and took approximately 5 minutes.

**Model Inference** was conducted using vLLM[27]. Vision tasks were processed on a single NVIDIA A40 GPU (48GB) over 3 hours, while text-only tasks were run on two NVIDIA L40 GPUs (48GB each) for approximately 1.5 hours. Image representation extraction for Qwen2.5VL was also performed on a single A40 GPU and completed in roughly 2.5 hours. Static embeddings were computed in under 10 minutes on an L40 GPU.

**TAXOMPS**, RSA on unembedding layer vectors, contextualized representational similarity analysis, and PCA analysis were conducted on a single NVIDIA RTX6000 Ada (48GB) GPU, and took a total of 1 hour, 1.5 hours, 4 hours, and 1 hour, respectively. Representation extraction and **TAXOMPS** behavioral analyses were performed using the `minicons` library [41]. All plots were produced using the `ggplot2` library in the R programming language.

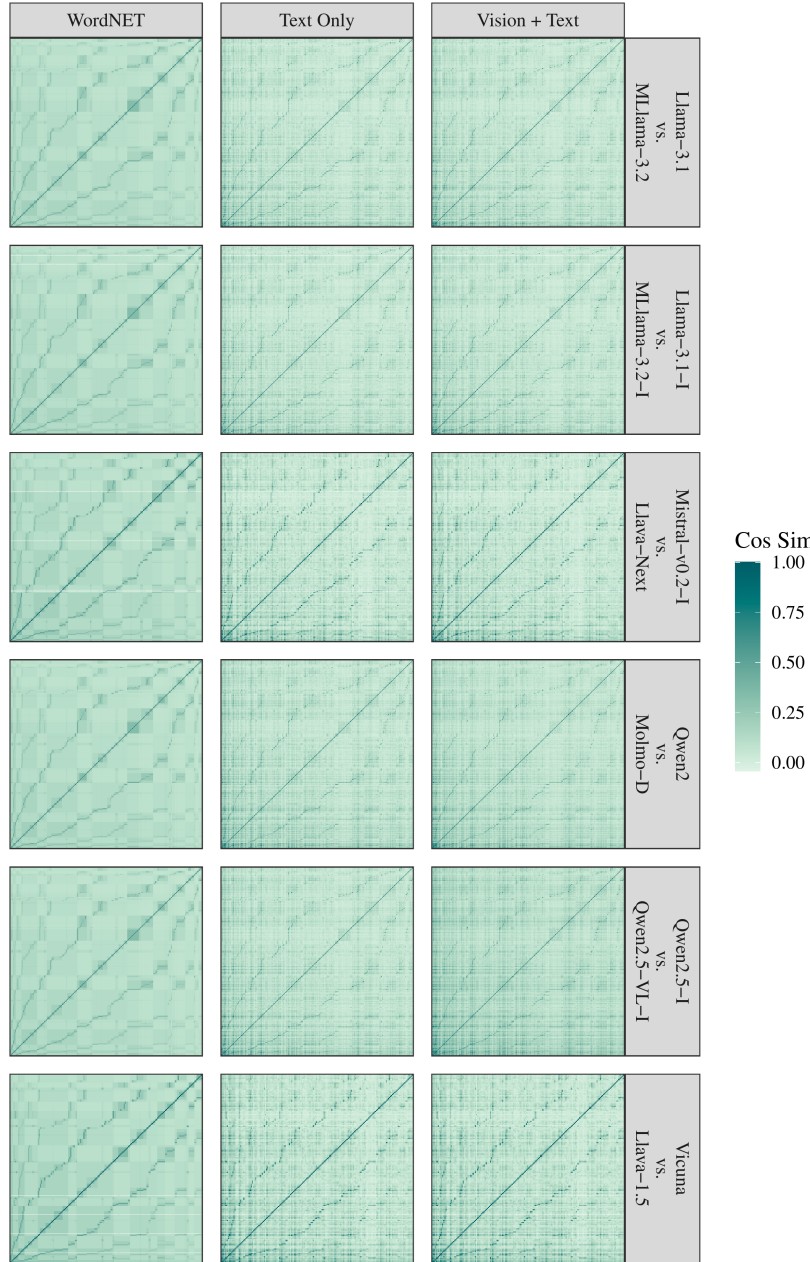

Figure 5: Pairwise similarities between concepts in WordNet, and the transformed unembedding spaces in Qwen2.5-I (LM) vs. Qwen2.5-VL-I (VLM) (computed using Park et al. [52]'s method), across all pairs.

We estimate that the total compute cost, including preliminary and unsuccessful experiments, is approximately **3x** the sum of the runtimes reported above.

## J License Information

The original GQA dataset was released under CC BY 4.0 and we downloaded the dataset from https://cs.stanford.edu/people/dorarad/gqa/download.html. We follow this and release **TaxonomiGQA** under the same license, CC BY 4.0.

