# OpenReview forum: "Vision-and-Language Training Helps Deploy Taxonomic Knowledge but Does Not Fundamentally Alter It"
_NeurIPS.cc/2025/Conference — NeurIPS 2025 poster_

### Official Review · Reviewer_PQ2W · 2025-06-19

**Clarity:** 3
**Significance:** 2
**Originality:** 3
**Rating:** 4
**Confidence:** 3

**Summary:**

This paper investigates the impact of vision-and-language (VL) training on the linguistic representations of language models (LMs).  The authors hypothesize that VL training may significantly affect lexical-conceptual knowledge, particularly its taxonomic organization (e.g., understanding that a cat is an animal). They created TaxonomiGQA, a text-only question-answering dataset derived from the GQA dataset. TaxonomiGQA requires understanding the taxonomic relationships between concepts, making it a suitable tool for evaluating the impact of VL training on taxonomic knowledge.

**Questions:**

- The study focuses on a specific type of knowledge (taxonomic) and a limited set of VLM-LM pairs. How can you extend the research to explore the impact of VL training on other types of knowledge and a broader range of language models?
- The study uses a logistic regression model to predict VLM performance based on contextual representation similarity. Are there any other better solutions?
- Are there any similar analysing works on VLM taxonomic knowledge? Can you compare your work with those?

**Ethical Concerns:**

["NO or VERY MINOR ethics concerns only"]

**Final Justification:**

The response provides some detailed explaination and some of my concerns are solved. However, justifying taxonomic focus doesn’t resolve the narrow scope; no evidence shows findings extend to other knowledge types. So I would like to keep my scores.

**Limitations:**

yes

**Quality:**

3

**Strengths And Weaknesses:**

- The hypotheses and anaylsing processes are intriguing.
- The paper is well-written and clear.
- The results are primarily focused on a specific type of knowledge (taxonomic), and it is unclear how generalizable the findings are to other types of knowledge.
- The use of TaxonomiGQA to evaluate the impact of VL training on taxonomic knowledge is a novel approach.

---

> ### Author Rebuttal · Authors · 2025-07-31
>
> Thank you for taking the time to offer us feedback. We appreciate your positive evaluation regarding the clarity of the paper, the hypotheses, and the overall framework. Below, we address the weaknesses you identified and questions you asked per topic, and hope our responses help resolve your concerns.
>
> **On generalizing from taxonomic knowledge to other types of knowledge:** Thank you for bringing up this point, something that Reviewer uJ4t also brought up. As they (Reviewer uJ4t) pointed out, focusing on one phenomenon allows us to perform more in-depth exploration/analysis. In addition, taxonomic knowledge is fundamental to semantic cognition research [1-5], and there has been a long precedence of this type of knowledge in ML – with datasets like ImageNet being built on top of the wordnet taxonomy. Secondly, prior attempts to assess the benefits of vision-and-language (VL) training for general language tasks have largely yielded negative results [6-11]. This suggests that broader generalization beyond taxonomic knowledge, so far, is a challenging endeavor.
>
> This is not to say that we disagree with the fact that the study is limited in scope – we appreciate the suggestions to go beyond taxonomic knowledge and we will be taking our current work as a blueprint to perform large scale behavioral benchmarking in the future, including but not limited to a text-only spatial reasoning dataset.
>
>
> **On the usage of logistic regression:** We intentionally chose logistic regression because it is well-suited for interpretability: the odds ratios are effect sizes that directly quantify how much contextual similarity (independent variable) contributes to correctness of answers (dependent variable). While more complex models (e.g., MLPs, SAEs, or even LLMs) might yield higher predictive accuracy, they would obscure the clear, coefficient-based interpretation critical to our analysis. Our goal was not to optimize predictive performance, but to demonstrate whether increased contextual similarity between concepts is systematically associated with better task performance, a conclusion that logistic regression enables us to draw in a transparent manner. This has been standard practice in psycholinguistics, among other disciplines.
>
> **On whether similar works exist and if we could compare against them:** Taxonomic knowledge has certainly been of interest to the community, both in terms of evaluating text-only models as well as VLMs. To the best of our knowledge, no former work has evaluated the impact of VL-training on LMs’ taxonomic/category knowledge using minimal-pair VLM-LM architectures. Furthermore, since this minimal pair comparison involves a text only task, we cannot use datasets that have only been used to evaluate VLMs. A related point was brought up by reviewer p9UG, and this is part of our response to them relevant to your comment: There are three classes of taxonomically motivated datasets that we could have compared against, enumerated below:
>
> 1) Word-prediction datasets like the one by Hanna and Mareček [12] use cloze-prediction tasks to test models like BERT. While useful, its design closely resembles the positive examples of TAXOMPs and does not add substantial new evaluation challenges.
>
> 2) NLI datasets like those used in Geiger et al. [13] focus on lexical entailment (inferred using hypernymy) but lack negative samples, which makes it less suitable for robust evaluations.
>
> 3) Property inheritance testing datasets like in Rodriguez, Mueller, Misra [14] involve attributing a novel, nonsense property (e.g., is daxable) to concepts given that their parents are attributed with it – e.g., Given that all birds are daxable, is it true that robins are daxable? -> Yes/No. This dataset does include a robustness check in the form of a single negative sample per concept (we have four), and even includes a larger set of categories (2016 positive samples in total, spanning 44 superordinate categories and 1281 subordinate categories). Therefore, it proves to be a more suitable candidate compared to 1) and 2) for (additionally) analyzing deployment of taxonomic knowledge. We tested our model pairs on this dataset (with close to no prompt-tuning), and found largely similar qualitative trends, as shown in the following table. We will include this in the appendix as supplementary evidence that supports our conclusions. We appreciate the reviewer’s suggestion and hope this addresses the concern.
>
> | LM-Model Name                           | LM-Score | VLM-Model Name        | VLM-Score |
> |----------------------------------------|----------|------------------------|-----------|
> | lm_Qwen_Qwen2.5_7B_Instruct            | 0.78     | vlm_text_qwen2.5VL     | **0.83**      |
> | lm_Qwen_Qwen2_7B                       | 0.60     | vlm_text_molmo_D       | **0.81**      |
> | lm_Qwen_Qwen2_7B_Instruct              | **0.84**     | vlm_text_llava_ov      | 0.81      |
> | lm_mistralai_Mistral_7B_Instruct_v0.2 | 0.69     | vlm_text_llava_next    | **0.78**      |
> | lm_lmsys_vicuna_7b_v1.5               | 0.50     | vlm_text_llava         | **0.74**      |
> | lm_meta_llama_Llama_3.1_8B            | 0.50     | vlm_text_mllama        | **0.68**      |
>
> **References:**
>
> [1] Murphy, G. (2002). The big book of concepts. MIT press.
>
> [2] Collins, A. M., & Quillian, M. R. (1969). Retrieval time from semantic memory. Journal of verbal learning and verbal behavior, 8(2), 240-247.
>
> [3] Keil, F. C. (1992). Concepts, kinds, and cognitive development. mit Press.
>
> [4] Rosch, E. (1975). Cognitive representations of semantic categories. Journal of experimental psychology: General, 104(3), 192.
>
> [5] Xu, F., & Tenenbaum, J. B. (2007). Word learning as Bayesian inference. Psychological review, 114(2), 245.
>
> [6] Yun, T., Sun, C., & Pavlick, E. (2021). Does vision-and-language pretraining improve lexical grounding? EMNLP
>
> [7] Amariucai, T., & Warstadt, A. (2024). Acquiring linguistic knowledge from multimodal input. BabyLM.
>
> [8] Wang, W., Vong, W. K., Kim, N., & Lake, B. M. (2023). Finding structure in one child's linguistic experience. Cognitive science.
>
> [9] Deitke et al. (2024). Molmo and Pixmo: Open weights and open data for state-of-the-art multimodal models.
>
> [10] Iki, T. & Aizawa, A. (2021). Effect of visual extensions on natural language understanding in vision-and-language models. EMNLP
>
> [11] Madasu, A. & Lal, V. (2023). Is multimodal vision supervision beneficial to language? CVPR
>
> [12] Hanna, M., & Mareček, D. (2021). Analyzing BERT’s knowledge of hypernymy via prompting. BlackboxNLP.
>
> [13] Geiger, A., Richardson, K., & Potts, C. (2020). Neural natural language inference models partially embed theories of lexical entailment and negation. arXiv preprint arXiv:2004.14623.
>
> [14] Rodriguez, J. D., Mueller, A., & Misra, K. (2025). Characterizing the Role of Similarity in the Property Inferences of Language Models. NAACL

---

> > ### Comment · Reviewer_PQ2W · 2025-08-03
> > **I have read the rebuttal**
> >
> > Thanks for the response. It provides some detailed explaination and some of my concerns are solved. However, justifying taxonomic focus doesn’t resolve the narrow scope; no evidence shows findings extend to other knowledge types. So I would like to keep my scores.

---

> > > ### Author Response · Authors · 2025-08-05
> > > **Thanks for engaging!**
> > >
> > > Thanks for engaging with our response! We would like to reiterate that there have been many prior explorations that tried to investigate the same broad question as ours, and found vision-and-language training to show no improvement over language-alone training (sometimes even showing a decrease in performance) -- see refs 6–11 in our previous response. So for domains that prior work has investigated, additional visual modality does not provide much benefit. While our findings focused on taxonomic knowledge could be interpreted as "narrow scope" and may not extend to previously investigated domains, we in fact believe this context makes our finding even more interesting, since ours is the first case demonstrating that V&L training results in a noticeable gain in performance. Furthermore, the fact that the domain in which this happens is clearly identifiable is also beneficial for future research---tiny gains in very general domains like GLUE or LM perplexity, as in FLAVA and [8], makes it difficult to conduct targeted investigations about the underlying reason for the gains. We would like to urge you to consider evaluating our work in this broader context -- we think it is exciting that we found *any* tangible area of improvement attributable to additional visual modality in light of previous unsuccessful attempts!
> > >
> > > Of course, we agree with you that it would be great to see whether there are other areas outside of taxonomic knowledge our findings hold. If you had any other specific domains in mind, with a readily available evaluation dataset that is as controlled as our TaxonomiGQA, please let us know, and we would be happy to extend our experiments.

---

### Official Review · Reviewer_uJ4t · 2025-07-03

**Clarity:** 4
**Significance:** 2
**Originality:** 2
**Rating:** 5
**Confidence:** 3

**Summary:**

This paper investigates how vision-and-language (VL) training affects the taxonomic knowledge within language models (LMs). The authors introduce TaxonomiGQA, a novel text-only question-answering dataset designed to evaluate taxonomic understanding. By comparing LMs with their corresponding VL-trained versions (VLMs), the study demonstrates that VLMs outperform their text-only counterparts. The authors present a series of behavioral and representational analyses suggesting this advantage stems not from a fundamental alteration of the models' stored taxonomic knowledge, but from an enhanced ability to deploy this knowledge within a specific task context.

**Questions:**

1. The central claim is that VL training enhances knowledge deployment rather than the knowledge base itself. How might this hypothesis be tested beyond taxonomy? For example, could a similar methodology be applied to evaluate a model's understanding of spatial relations, by creating minimal-pair questions where only the relation changes? Demonstrating the effect's generality could help to strengthen the paper's impact.
2. The manual filtering of the taxonomy is a potential source of bias. Could you elaborate on the specific criteria used to exclude concepts as "too abstract"? Providing a more formal definition or even examples of rejected concepts in the appendix would improve the transparency of the dataset's construction.

**Ethical Concerns:**

["NO or VERY MINOR ethics concerns only"]

**Final Justification:**

The authors' rebuttal successfully addressed my main concerns. They provided a compelling justification for the paper's scope and novelty, and their detailed explanation of the data filtering process resolved my previous concerns. With these key issues clarified, I believe the paper makes a solid contribution, and I recommend acceptance.

**Limitations:**

yes

**Quality:**

3

**Strengths And Weaknesses:**

**Strengths:**
- **Significance and Originality**: The paper addresses a key question in multimodal AI. By focusing on how VL training influences the deployment of knowledge acquired during pre-training, the work offers a new perspective on language grounding in multimodal models and how this influences downstream reasoning.
- **Methods**: The creation of the TaxonomiGQA dataset is a valuable contribution for probing conceptual knowledge in taxonomic reasoning. The systematic testing of two explicit hypotheses (H1 and H2) with a suite of targeted behavioral and representational analyses (e.g., RSA, PCA) lends strong support to the conclusions.
- **Clarity**: The paper is well-written and logically structured. The motivation, hypotheses, and experimental design are presented clearly. The figures and tables are informative and effectively support the paper's claims.

**Weaknesses:**
- **Limited Scope**: The study's primary focus is on taxonomic knowledge. While this allows for a deep and controlled analysis, it's unclear whether these differences generalize to other task domains.
- **Data filtering details**: The creation of the reference taxonomy involved manual filtering of abstract concepts and verification of attributes. A more detailed appendix with the specific criteria for these manual steps would enhance the work's reproducibility.
- **Ambiguity of "Deployment"**: The paper argues for an improvement in "deployment," supported by evidence from contextualized representations. However, the precise mechanisms of this improved deployment remain underspecified. The link to visual similarity is a valuable first step but is acknowledged as preliminary and not a complete explanation.

---

> ### Author Rebuttal · Authors · 2025-07-31
>
> Thanks for taking the time to offer us feedback! We appreciate your positive comments regarding our controlled setup, the clarity of our hypotheses, and the representational depth of our analyses. Below, we address the weaknesses you identified and questions you asked clustered by topic, and hope our responses help resolve your concerns.
>
> **On limited scope and generalization of results beyond taxonomic knowledge:** Thank you for raising this concern. As you have pointed out, focusing on one phenomenon allows us to perform more in-depth exploration/analysis. In addition, taxonomic knowledge is fundamental to semantic cognition research [1-5], and there has been a long precedence of this type of knowledge in ML – with datasets like ImageNet being built on top of the wordnet taxonomy. Secondly, prior attempts to assess the benefits of vision-and-language (VL) training for general language tasks have largely yielded negative results [6-11]. This suggests that broader generalization beyond taxonomic knowledge, so far, is a challenging endeavor.
>
> This is not to say that we disagree with the fact that the study is limited in scope – we appreciate the suggestions to go beyond taxonomic knowledge and we will be taking our current work as a blueprint to perform large scale behavioral benchmarking in the future, including but not limited to a text-only spatial reasoning dataset.
>
> **On data filtering details:** We appreciate you raising this concern, and apologize for lack of clarity. As mentioned in the manuscript, this was done over multiple stages. Before manual annotation and removal of specific chains, we identified 52 highly abstract concepts (e.g., entity, physical entity, whole, conveyance, act, etc.) to remove from our chains right away – these are listed in the “GQA Hierarchies First Pass.ipynb” notebook under `notebooks/` in our code. After generating the hypernym chains, we then resorted to further manual inspection to find and remove/replace cases with “non-ideal” categorizations – by which we meant either that it was not the canonical category of the object (i.e., bubble being categorized as circle) or that there were chains that only contained abstract elements initially missed from our filtering. We found a total of 611 such cases, out of which we were able to “fix” 296 cases by simply querying wordnet to find alternative chains, leaving us with 315 unfixable cases, which were removed (as stated in the paper). These annotations can be found in `data/gqa_entities/noun-hypernymy-paths-annotated.csv` – with “1” indicating unfixable cases, “2” indicated replaced cases, and “3” indicating cases we were not sure about (N=3) which we decided to remove. All manual annotation was done by an expert in the team with 7+ years experience with semantic knowledge literature. We will disclose these additional details in the final version of our paper, and once again thank you for bringing this to our attention. In the context of attributes: we remove ones that introduce non-standard property attribution, such as “sitting airplanes”, or “happy trees” or “fluffy apple” or “swimming flowers”. We will aim to report a sample of the statistics for such cases in the final version.
>
> **On underspecification of deployment mechanisms:** This is indeed an important question that has not escaped our minds. First, by deployment, we mean the application of knowledge in contexts that implicitly require this knowledge to be recruited. This is what we measure using TaxonomiGQA. We differentiate this from explicit knowledge and representation/storage, which we measure using TAXOMPS (explicit questions about category membership) and representational analyses as part of H1 in the paper. Next, getting at the underlying mechanisms will require a whole host of substantially different tools (such as training data attribution or controlled training ablations) that are a bit out of scope of this work. This is because the meaningful usage of those tools or pursuing that research question depends on the prerequisite finding of a task where vision and language does have a positive impact on language only-task, which has so far been elusive, until this work.
>
> **References:**
>
> [1] Murphy, G. (2002). The big book of concepts. MIT press.
>
> [2] Collins, A. M., & Quillian, M. R. (1969). Retrieval time from semantic memory. Journal of verbal learning and verbal behavior.
>
> [3] Keil, F. C. (1992). Concepts, kinds, and cognitive development.
>
> [4] Rosch, E. (1975). Cognitive representations of semantic categories. Journal of experimental psychology: General.
> [5] Xu, F., & Tenenbaum, J. B. (2007). Word learning as Bayesian inference. Psychological review, 114(2), 245.
>
> [6] Yun, T., Sun, C., & Pavlick, E. (2021). Does vision-and-language pretraining improve lexical grounding? EMNLP
>
> [7] Amariucai, T., & Warstadt, A. (2024). Acquiring linguistic knowledge from multimodal input. BabyLM.
>
> [8] Wang, W., Vong, W. K., Kim, N., & Lake, B. M. (2023). Finding structure in one child's linguistic experience. Cognitive science.
>
> [9] Deitke et al. (2024). Molmo and Pixmo: Open weights and open data for state-of-the-art multimodal models.
>
> [10] Iki, T. & Aizawa, A. (2021). Effect of visual extensions on natural language understanding in vision-and-language models. EMNLP
>
> [11] Madasu, A. & Lal, V. (2023). Is multimodal vision supervision beneficial to language? CVPR

---

> > ### Comment · Reviewer_uJ4t · 2025-08-05
> >
> > Thank you for the detailed and thoughtful rebuttal, which successfully addressed my main concerns. I now better understand the authors focus on taxonomic knowledge, and the novelty of their contribution relative to other work that has struggled to demonstrate the benefits of multimodal training on text-only reasoning. The additional details on your data filtering process resolve my concerns about transparency, and your clarification distinguishing “knowledge deployment” from “storage” was helpful. Your response has resolved my core concerns, and I will increase my score accordingly.

---

### Official Review · Reviewer_p9UG · 2025-07-05

**Clarity:** 3
**Significance:** 2
**Originality:** 3
**Rating:** 3
**Confidence:** 3

**Summary:**

This paper introduces a new text-only question answering dataset, TaxonomiGQA, designed to evaluate models' taxonomic understanding capabilities. It reveals that Vision-Language Models (VLMs) demonstrate stronger taxonomic performance compared to their original Language Models (LMs). The authors propose two hypotheses and provide experimental evidence showing that visual training helps enhance the models' ability to utilize taxonomic knowledge, rather than directly increasing the amount of such knowledge.

**Questions:**

Why didn’t the authors consider using existing taxonomic datasets for evaluation? In my view, doing so would provide a more comprehensive perspective on the observed phenomenon.

**Ethical Concerns:**

["NO or VERY MINOR ethics concerns only"]

**Final Justification:**

Thanks for answering my question, I decided to keep my original score.

**Limitations:**

yes

**Paper Formatting Concerns:**

No concerns

**Quality:**

3

**Strengths And Weaknesses:**

**Strengths:**

1. The paper introduces a new text-only QA dataset, **TaxonomiGQA**, for evaluating models' taxonomic understanding, along with a novel metric tailored for measurement.
2. The paper reports an interesting finding: VLM training helps improve the model’s ability to utilize taxonomic knowledge.
3. The paper proposes two plausible hypotheses to explain the observed phenomenon and provides thorough experimental validation.

**Weaknesses:**

1. The study primarily evaluates models’ taxonomic knowledge using **TaxonomiGQA** , which is built upon the fixed hierarchical structure of WordNet. However, this structure may not fully capture the dynamic and polysemous nature of real-world concepts.
2. Given that many of the VLMs you use were trained on GQA, evaluating them using TaxonomiGQA—which is constructed based on GQA—raises concerns about fairness, even though the paper mentions that VLMs do not perform significantly above chance level. Therefore, constructing the benchmark on other text-only datasets would make the evaluation more convincing.
3. In my view, the phenomenon and explanation presented in the paper lack substantial significance and strong novelty, and they do not address a concrete problem

---

> ### Author Rebuttal · Authors · 2025-07-31
>
> Thank you for taking the time to offer us feedback. We are glad that you recognized the effort involved in the construction of the TaxonomiGQA dataset, found our evaluation metrics novel, and viewed our hypotheses on VLM training and taxonomic knowledge deployment (as well as their validation) as thorough and well-motivated. Below, we address the weaknesses you identified clubbed by topic, and hope our responses help resolve your concerns.
>
> **On the fixed structure of WordNet and potentially ignoring the “dynamic” and “polysemous nature of real world concepts”:** Thank you for bringing this up. We agree that polysemy and the context-sensitive nature of concepts is important. We also agree that it can be difficult to draw a sharp line even between physical entities, let alone the abstract ones such as “law”, “emotion”, or “right”, given the dynamic and context-sensitive nature of concepts. Nonetheless, we believe WordNet offers a reasonable approximation of human-like taxonomies and remains a widely used resource for taxonomy-related tasks throughout computational linguistics [1-3, for example]. Moreover, even within this constrained, fixed-structure setting, models still struggle to demonstrate robust taxonomic understanding, underscoring the value of using WordNet-based benchmarks as a good starting point for taxonomic evaluation. More importantly, we’re also not aware of any resources that can faithfully facilitate research into dynamic, polysemy aware taxonomic/category organization. If a more realistic taxonomy dataset becomes available, we would be glad to extend our experiments to it. We also agree that developing context-sensitive and usage-driven concept structures is a valuable direction for future research.
>
> **On the potential “unfairness” of using GQA questions:** Thank you for raising this concern. Although TaxonomiGQA builds on top of GQA’s base questions, there are critical elements to our design that mitigate this risk. First, TaxonomiGQA is a text-only task – each image is converted into a scene description, and since the image is discarded in lieu of this long-form description, this is no longer the same task as the original GQA. Next, the original GQA questions do not contain hypernym substitutions or negative samples, by including them, we again deviate substantially from the original task (many of them introduce label changes too). Third, we have an important constraint in place for the conditional and HCA metrics – that each respective model must correctly answer the base question (i.e., the question that appeared verbatim in GQA). So in this sense, no model has a particularly unfair advantage. Finally, we had foreseen this concern, and ran a question-only control to see if VLMs have an unfair advantage – the inputs are now identical to the original GQA task’s text content. We found that models are substantially worse in this setting, being closer to chance (50%) – results can be found in Table 4, appendix D).
>
> **On the lack of “strong novelty” and failure to “address a concrete problem”:** While we appreciate you bringing up this concern, we will have to respectfully disagree with it. Taxonomic knowledge is one of the most prominent types of knowledge studied in the semantic cognition literature [5-12] – it is not totally clear what the origins of its development are, and what kinds of features and input signals are central to learning it. So in this sense, we believe taxonomic knowledge/hypernymy to be a fundamental lexical semantic concept worth studying in systems that seemingly demonstrate linguistic competence. In the context of novelty, there is an abundance of investigations trying to see whether multimodal training leads to improvements on language-only task performance of an LM, all of them finding negative results [13-18]. To the best of our knowledge, ours is the first setting where we can conclude that V-L training (a subset of all kinds of multi-modal training) has brought about a substantial improvement in an identifiable domain (e.g., Wang et al.’s [15] marginal improvement in perplexity in VLMs were not attributed to specific domains), and that it is nuanced (i.e., no real change in representation but non-zero change in deployment) We hope that this discussion presents compelling evidence about the concreteness of the domain we are hoping to work on, and the novelty of our findings.
>
> **On not using existing taxonomic datasets:** We want to clarify that we did, in fact, incorporate multiple existing taxonomic datasets into our evaluation, in addition to the TaxonomiGQA dataset we introduce. First, the TAXOMPs dataset which we created using the GQA entities, which behaviorally allows us to test explicit taxonomic membership. Then, one of our representational analyses which uses the methodology from Park et al. [9] and uses quite a large subset of the wordnet taxonomy – larger than the set of GQA concepts, to see the general geometry of hierarchical representations within the embeddings of VLM-LM pairs. Outside of the datasets we used, there are three kinds of taxonomically motivated datasets that we could have used, we enumerate them below:
>
> 1) Word-prediction datasets like the one by Hanna and Mareček [10] use cloze-prediction tasks to test models like BERT. While useful, its design closely resembles the positive examples of TAXOMPs and does not add substantial new evaluation challenges.
>
> 2) NLI datasets like those used in Geiger et al. [11] focus on lexical entailment (inferred using hypernymy) but lack negative samples, which makes it less suitable for robust evaluations.
>
> 3) Property inheritance testing datasets like in Rodriguez, Mueller, Misra [12] involve attributing a novel, nonsense property (e.g., is daxable) to concepts given that their parents are attributed with it – e.g., Given that all birds are daxable, is it true that robins are daxable? -> Yes/No. This dataset does include a robustness check in the form of a single negative sample per concept (we have four), and even includes a larger set of categories (2016 positive samples in total, spanning 44 superordinate categories and 1281 subordinate categories). Therefore, it proves to be a more suitable candidate compared to 1) and 2) for (additionally) analyzing deployment of taxonomic knowledge. We tested our model pairs on this dataset (with close to no prompt-tuning), and found largely similar qualitative trends, as shown in the following table. We will include this in the appendix as supplementary evidence that supports our conclusions. We appreciate the reviewer’s suggestion and hope this addresses the concern.
>
> | LM-Model Name                           | LM-Score | VLM-Model Name        | VLM-Score |
> |----------------------------------------|----------|------------------------|-----------|
> | lm_Qwen_Qwen2.5_7B_Instruct            | 0.78     | vlm_text_qwen2.5VL     | **0.83**      |
> | lm_Qwen_Qwen2_7B                       | 0.60     | vlm_text_molmo_D       | **0.81**      |
> | lm_Qwen_Qwen2_7B_Instruct              | **0.84**     | vlm_text_llava_ov      | 0.81      |
> | lm_mistralai_Mistral_7B_Instruct_v0.2 | 0.69     | vlm_text_llava_next    | **0.78**      |
> | lm_lmsys_vicuna_7b_v1.5               | 0.50     | vlm_text_llava         | **0.74**      |
> | lm_meta_llama_Llama_3.1_8B            | 0.50     | vlm_text_mllama        | **0.68**      |
>
> **References:**
>
> [1] Cho, Y., Rodriguez, J. D., Gao, Y., & Erk, K. (2020). Leveraging WordNet paths for neural hypernym prediction. COLING
>
> [2] Misra, K., Rayz, J., & Ettinger, A. (2023). COMPS: Conceptual Minimal Pair Sentences for testing Robust Property Knowledge and its Inheritance in Pre-trained Language Models. EACL
>
> [3] Moskvoretskii, V., Panchenko, A., & Nikishina, I. (2024). Are large language models good at lexical semantics? a case of taxonomy learning. LREC-COLING 2024.
>
> [4] Murphy, G. (2002). The big book of concepts.
>
> [5] Collins, A. M., & Quillian, M. R. (1969). Retrieval time from semantic memory. Journal of verbal learning and verbal behavior
>
> [6] Keil, F. C. (1992). Concepts, kinds, and cognitive development.
>
> [7] Rosch, E. (1975). Cognitive representations of semantic categories. Journal of experimental psychology: General, 104(3), 192.
>
> [8] Xu, F., & Tenenbaum, J. B. (2007). Word learning as Bayesian inference. Psychological review
>
> [9] Park, K., Choe, Y. J., Jiang, Y., & Veitch, V. (2024) The Geometry of Categorical and Hierarchical Concepts in Large Language Models. ICLR.
>
> [10] Hanna, M., & Mareček, D. (2021). Analyzing BERT’s knowledge of hypernymy via prompting. BlackboxNLP.
>
> [11] Geiger, A., Richardson, K., & Potts, C. (2020). Neural natural language inference models partially embed theories of lexical entailment and negation. arXiv preprint arXiv:2004.14623.
>
> [12] Rodriguez, J. D., Mueller, A., & Misra, K. (2025). Characterizing the Role of Similarity in the Property Inferences of Language Models. NAACL.
>
> [13] Yun, T., Sun, C., & Pavlick, E. (2021). Does vision-and-language pretraining improve lexical grounding? EMNLP
>
> [14] Amariucai, T., & Warstadt, A. (2024). Acquiring linguistic knowledge from multimodal input. BabyLM.
>
> [15] Wang, W., Vong, W. K., Kim, N., & Lake, B. M. (2023). Finding structure in one child's linguistic experience. Cognitive science.
>
> [16] Deitke et al. (2024). Molmo and Pixmo: Open weights and open data for state-of-the-art multimodal models.
>
> [17] Iki, T. & Aizawa, A. (2021). Effect of visual extensions on natural language understanding in vision-and-language models. EMNLP
>
> [18] Avinash Madasu and Vasudev Lal (2023). Is multimodal vision supervision beneficial to language? CVPR

---

### Official Review · Reviewer_SKmA · 2025-07-12

**Clarity:** 3
**Significance:** 3
**Originality:** 3
**Rating:** 5
**Confidence:** 3

**Summary:**

This paper investigates how vision-and-language training affects the linguistic knowledge of language models (LMs), focusing on taxonomic relationships, by comparing several "minimal pairs" of a text-only LM and its VL-trained counterpart (VLM). Two competing hypotheses the paper tests are: (H1) that VL training fundamentally alters a model's stored taxonomic knowledge, and (H2) that it improves the model's ability to deploy this knowledge in context. The paper introduces two novel datasets: TAXOMPS, a direct knowledge elicitation task (e.g., "Is a cat an animal?"), they find little difference between LMs and VLMs, providing evidence against H1. Using TaxonomiGQA, a text-based QA task requiring contextual inference (e.g., given a text about a 'cat', answer if there is an 'animal'), they find that VLMs often outperform LMs, which supports H2.

**Questions:**

- Could the authors elaborate on the motivation for focusing on taxonomic knowledge? While it's a core part of conceptual structure, one might argue that other aspects (e.g. spatial prepositions) are more directly linked to visual input. What is the specific hypothesis about the role of vision in shaping abstract hierarchical knowledge that makes taxonomy a more interesting test case than these other aspects?
- The Vicuna vs. Llava-1.5 pair is noted as a consistent exception, with the LM performing on par with or better than the VLM (Figure 2). Do the authors have any specific hypotheses for why this pair deviates from the general trend?
- One of the model pairs, Llama-3.1 vs MLlama-3.2, has a notable size difference (8B vs 11B). It seems like this difference in parameter count might be a reason why this specific pair is an outlier in the TAXOMPS task but not in so much in TaxonomiGQA. Does the authors have a hypothesis regarding why that is?

**Ethical Concerns:**

["NO or VERY MINOR ethics concerns only"]

**Final Justification:**

I appreciate the authors' detailed response. In particular, I found the discussion on the importance of distinguishing between existing knowledge and its deployment to be valuable. I’m satisfied with their explanation and am therefore increasing my score, assuming these clarifications will be incorporated into the revision.

**Limitations:**

- The authors have addressed the primary limitations of their work, identifying that their analyses establish correlation, not causation, and that achieving causal evidence is a significant challenge. They also acknowledge the difficulties of finding perfectly "minimal" model pairs.

**Quality:**

3

**Strengths And Weaknesses:**

##### Strengths
- Using TAXOMPS to test for knowledge possession and TaxonomiGQA to test for knowledge deployment allows for a clean disentanglement of the paper's two central hypotheses.
- The use of negative samples guards against superficial model behavior, and the "Conditional Accuracy" metric is tailored to measure the specific hierarchical inference, which is critical to the research question.
- The paper presents a very thorough analysis, combining behavioral tests with multiple layers of representational analysis (static embeddings, contextual embeddings, full-sentence representations).
##### Weaknesses
- The main finding that VL training primarily helps deploy existing knowledge rather than fundamentally altering it is a non-obvious insight but why knowing this is important for the community is somewhat unclear.
- Not enough expalanation for why some model pairs don't follow the trend (e.g. Vicuna vs. Llava-1.5 is an outlier in TaxonomiGQA experiment, and Qwen2-I vs. Llava-OV is another outlier in TAXOMPS experiment)
- By nature, the analyses are correlational. Although the paper is upfront about this, it remains an inherent weakness of this style of analysis that it cannot establish causality.

---

> ### Author Rebuttal · Authors · 2025-07-31
>
> Thank you for taking the time to give us feedback. We appreciate your positive comments regarding our dataset design, choice of metric, inclusion of negative samples, and our analyses. Below, we address the weaknesses you identified and questions you asked grouped within topics, and hope our responses help resolve your concerns.
>
> **On the importance of distinction between existing knowledge vs. its deployment:** We see two specific reasons why this distinction could be important: first, storing/representing knowledge is different from learning their functional consequences [1]. For instance, a model may as well robustly represent category information (robins are birds) but might not deploy this correctly in contexts that recruit this knowledge - we see TaxonomiGQA as one such context. The practical benefit of teasing this apart is knowing where the performance bottleneck lies, and this could inform decisions about training data selection (e.g., if we were to collect additional (post)training data, should we include more encyclopedic knowledge or diverse contexts in which such knowledge is deployed?) and motivate solutions such as transferring task vectors [2] from models that are better at deployment, for instance. Next, finding that there is a distinction between representation of knowledge vs. its deployment and that it could depend upon the kind of supervision received by a neural network model can be informative for cognitively inclined researchers who want to draw appropriate conclusions about the role of language vs. extralinguistic information. For instance, the platonic representational hypothesis [3] roughly suggests that models trained on a large amounts of data in the world learn similar representations, even if their input modalities are completely different (text vs. vision); our findings suggests, to an extent, that while this might be possible, there *are* cases where combining information from multiple modalities can bring about non-trivial changes, not in the representations (of a certain kind – in our case taxonomies) but instead in terms of its deployment.
>
> **On outliers in results:** We appreciate you drawing attention to two main outliers in our results: first with the LM outperforming the VLM on the TaxonomiGQA task for the Vicuna-Llava pair; and the second with the VLM substantially outperforming the LM mllama-llama 3.1 pair on TAXOMPS. While we agree that understanding precisely what leads to these exceptions would be beneficial, we would like to point out that our main goal in this paper is about the effect of *V-L training*, holding the underlying architectures of the models constant. In this framing, our comparisons are meant to isolate the presence or absence of VL training. Explaining the behavior of a single outlier like Vicuna vs. LLaVA-1.5 would require controlled architectural ablations, which are beyond the scope of our current setup. A tentative conjecture one could make about Mllama 3.2 being substantially better than Llama 3.1 on the TAXOMPs task is that the Mllama 3.2 model introduces 2 additional elements to the LLama 3.1 model: 1) 3B additional parameters due to cross-model attention; and 2) RLHF based instruction-tuning, despite not saying “Instruct” in the name (as reported in the HuggingFace documentation of the model). Since the Mllama 3.2-Instruct vs. Llama 3.1-Instruct pair does not show this exceptional behavior, 2) is perhaps the more plausible explanation for this exceptional behavior.
>
> **On the limitations of correlational analyses:** We agree this is a limitation and explicitly acknowledge it in the paper. Future work could use training data-centric interventions to ask causal questions, for instance by manipulating training data during VL training. However, given computational constraints (such experiments would entail multiple pretraining runs), we believe our correlational analyses and hypothesis testing are a meaningful first step.
>
> **References:**
>
> [1] Murphy, G. (2002). The big book of concepts. MIT press.
>
> [2] Hendel, R., Geva, M., & Globerson, A. (2023). In-context learning creates task vectors.
>
> [3] Huh, M., Cheung, B., Wang, T., & Isola, P. (2024). The platonic representation hypothesis. arXiv preprint arXiv:2405.07987.

---

> > ### Comment · Reviewer_SKmA · 2025-08-05
> > **response**
> >
> > I appreciate the authors' detailed response. In particular, I found the discussion on the importance of distinguishing between existing knowledge and its deployment to be valuable. I’m satisfied with their explanation and am therefore increasing my score, assuming these clarifications will be incorporated into the revision.

---

### Note · Authors · 2025-08-15

We thank all reviewers for engaging with the rebuttals and appreciate the score increases. We hope this means that the concern about scope and value of the work has been sufficiently addressed by our responses, and hope our excitement about positive results in this space is shared.

---

### Decision · Program_Chairs · 2025-09-17

**Decision:**

Accept (poster)

**Comment:**

The paper received four expert reviews. The authors provided a rebuttal that attempted to address the concerns raised in the reviews. The reviewers read the rebuttal and engaged with the authors. After the rebuttal and discussion, the reviewers a bit dived with final recommendations of accept, borderline reject, accept, and borderline accept. The area chair reviewed the reviews, the paper, rebuttal, and discussion.  Overall the AC liked the contributions of the paper.  The main criticism was the limitation of the evaluation to being on TaxonimicGQA and also not addressing a concrete problem.  As this is a paper focusing on evaluation and understanding, it is inherently needed to focus on a more narrow task and isn't about solving a problem so much as surfacing it.

Ultimately, the area chair decided to accept the paper. Congratulations! Please see the reviews for feedback on the paper to revise the final version of your paper and include any items promised in your rebuttal.